# Ecosystem-Based Approaches to Bioenergy and the Need for Regenerative Supply Options for Africa

**Lalisa Duguma** *[ID]**, Esther Kamwilu** [ID]**, Peter A Minang , Judith Nzyoka and Kennedy Muthee**

World Agroforestry (ICRAF), UN Avenue, Gigiri, P.O. Box 30677, Nairobi 00100, Kenya; karimiesther5@gmail.com (E.K.); A.Minang@cgiar.org (P.A.M.); j.nzyoka@cgiar.org (J.N.); k.muthee@cgiar.org (K.M.)
*   Correspondence: L.A.DUGUMA@CGIAR.ORG

**Abstract:** Energy supply systems in the tropics and subtropics are marred with considerable negative impacts on ecosystems, for example, forest loss and habitat destruction. This document examines the role of ecosystems in household energy supply in Africa and explores pathways to ecosystem-based approaches to bioenergy generation by building on the regenerative economy concept. An ecosystem-based approach to bioenergy is an energy supply and utilization mechanism aimed at enhancing sustainable management of the sources of ecosystems with minimal trade-offs on/from other sectors directly linked to energy issues. Our analysis revealed that about 87% of energy supply to the population originated from agroecosystems and is challenged by the severe ecosystem degradation happening due to natural and anthropogenic factors. However, ecosystem restoration and effective use of agricultural residues could provide hope for making energy supply sustainable. Our analysis showed that restoring sparsely vegetated areas and degraded forest and savannahs, promotion of agroforestry in degraded agricultural lands, and use of agricultural residues could generate close to 71 billion gigajoules (GJ) of energy and provide sufficient energy for about 2.5 billion people if implemented in all potential areas identified. Ecosystem-based approaches to bioenergy along with a well-balanced involvement of sectors and industry actors coupled with knowledgeable management of the ecosystem could lead to beneficial outcomes for the society and environment.

**Keywords:** bioenergy; ecosystem-based approaches; Africa; restoration; regenerative economy; technology

## 1. Introduction

Current energy generation and raw material production schemes for energy significantly impact the habitability of our planet—from coal, which has contributed about 0.3 °C of the 1 °C temperature rise observed since 1800, to oil palm production destroying forest-dependent people's livelihoods and wildlife habitats [1]. The extractive nature of the global energy supply system has had its negative social consequences on the planet too. These negative effects are magnified by limited consideration of the environmental consequences, human population needs, and the degree of technological advancements. Duguma et al. [2] illustrated the prevailing vicious cycle of destruction due to lack of proper solutions to address household energy needs among rural communities. The authors demonstrated how energy extraction affects forest ecosystems, food security, climate change, biodiversity conservation, land resources management, and put societal pressure on women and girls.

To take one example, reservoirs for hydropower stations continuously suffer siltation due to soil erosion and poor land management in the watersheds. The subsequent effect of this degradation leads to the rising demand for wood fuel (i.e., charcoal and firewood) for communities that use electricity. This rising demand could generate destructive supply mechanisms that further deteriorate

the ecosystem leading to biodiversity loss and ecosystem collapse. Such are the characteristics of degenerating systems, where the elements of the system are leading to a vicious cycle of energy poverty and ecosystem degradation and further complicating finding solutions to the problem [3,4]. Discourses on how to solve the problem require understanding the system and identifying the right leverage points to enable significant positive change [5,6].

So far, efforts to abate the system-wide effects of energy supply schemes [2] have focused more on technical solutions (i.e., technological innovations such as energy-efficient systems development, nuclear power, solar energy harvesting, and others). However, the key inputs or sources of inputs for the technical solutions originate from the ecosystem. Hence, technology alone is insufficient to solve the negative impacts of energy systems. The major challenge with technical solutions to energy systems is the weak adoption levels in contrast to the high adoption [7,8], the feasibility of the technologies within the socio-cultural aspects of community to use the products [9]. Even for technologies with higher thermal efficiency, such as biogas technology with efficiency of 50–65% [10], adoption is limited due to affordability and lack of awareness [11,12].

Adoption alone is not sufficient as Ruiz-Mercado et al. [13] indicated. The authors emphasized that sustained use of the adopted cooking techniques is crucial; implying that the broader impact of energy supply systems cannot be solved through technological solutions alone. Instead, there is a need for complementary interventions to boost ecosystem capacity to cater to energy raw materials coupled with efficiency improvements both in production and during use. Hence, regenerative energy supply systems that deliver benefits to the whole system [14] are needed to limit the negative impacts arising from extractive energy uses that may lead to the collapse of the whole ecosystem.

Here we examine and elucidate the role that ecosystems play in energy supply. We then explore options for regenerative energy supply models that reduce impacts of the energy supply system and invest in the management of the ecosystems while making use of unused resources. The U.S. Energy Information Administration (EIA) [15] indicated that even by 2040, Africa's energy sources may not shift that significantly, in other words, biomass-based energy will still predominate. Besides technological advances, it is crucial to consider boosting the capacity for managing ecosystems to continue supplying biomass energy that the population may continue to use either as unprocessed input (e.g., firewood), semi-processed input (e.g., charcoal), or fully converted input (e.g., biomass electricity, biogas, hydropower, among others).

This paper presents a regenerative energy supply concept to move towards eco-friendly energy production options that increase the sustainability of the ecosystem. By examining and understanding the role that ecosystems play in energy supply, we explore options for regenerative energy supply models that reduce the negative effects of current energy supply models and capitalize on unused local bioenergy resources. We also propose judicious management of local ecosystems that considers the ecosystem itself, its possible future trajectories, and the social, ecological, and economic dynamics in and around it. This will aid in (1) framing proper and sustainable management strategies for energy supply sources and (2) reducing the negative environmental impacts arising from the use of ecosystems to source energy.

## 2. A Conceptual Framework: Towards Regenerative Energy Supply Options

Owen et al. [16] highlighted that despite developed countries pushing for biomass energy as a renewable low carbon energy option, in Africa biomass energy is viewed as a retrogressive and environmentally damaging energy option. Irrespective of this, biomass energy is likely to remain the key pillar of the energy systems on the continent. Hence, innovative options to produce, convert, and utilize it are key. The ecosystem concept and regenerative energy are therefore crucial for the energy discourse on the continent.

The article "Regenerative Economy" [17] emphasizes that unless current business practices change, it is difficult to have a healthy planet since our current economic development approaches are not strongly considerate of the environmental consequences the built economy is creating. It argues that

natural systems continued to survive to date because they are regenerative. The emphasis in the regenerative economy [17] is the concept of "system". There is a connection between every component of a system, and a system is as strong as its weakest link, which Meadows [6] calls a leveraging point. A system operates with sets of universal principles and patterns: efforts to build the economy should then be able to simulate these universal principles and patterns that otherwise may damage the system structure. Liu et al. [18] argue that Earth on its own is a system, and all other sub-systems operate under the bigger principles and patterns. The authors argue that as a system, every element is interdependent, and there is a connection between all the components. It is in this inter-relation that materials flow, and for one to understand and conserve a system, it is essential to understand the flows in each direction in specific components of the system and how each flow relates to the stability of other system components.

Among the key arguments in regenerative economy [19] is that wealth should be considered in a holistic manner and not simply as monetary value; hence, it should include economic, social, and environmental wealth. The emphasis on GDP of our current economic thinking has led the global economy to focus on economic growth without due consideration for its social and environmental consequences [20]. Among these consequences are child labor, environmental pollution, human exploitation, and biodiversity loss. Liu et al. [18] contend that global sustainability challenges are interconnected and should not be treated as separate pieces. The authors argue for a system integration that looks at holistic solutions within the coupled human and natural systems so that sustainable solutions to the significant global challenges can be found.

Energy is the fundamental input to drive development. The entire earth system depends on energy to function. In any aspect of life and natural system, there is an energy aspect, particularly the biogeochemical processes. At the lowest scales, for example, smallholder households, villages, districts, and even at the national level, the context of energy changes depending on the local situation and priorities. Nonetheless, the inherent feature in systems, namely, interconnectedness, still prevails. Consequently, energy should be contextualized as an element of the broader system and not as a stand-alone issue. Within the energy sector itself, there are several relations, interactions, and flows that need to be viewed as a subsystem of the larger earth system. Therefore, we advocate strongly for the use of the concept of energy system(s) rather than referring to energy as a single piece of the issue. Figure 1 shows how the different energy sources that directly depend on the ecosystem relate to the sectors that policymakers tend to use. The conceptual representation does not show indirect linkages.

If the ecosystem elements on which energy supply depends are not appropriately managed, energy poverty is going to cripple the ecosystem and the economy broadly. Once ecosystem degradation peaks, its habitability could significantly decline as its potential to supply goods and services on which life depends gradually diminish. This creates a series of strong arguments for ecosystem-based approaches to energy production as formulated here.

Argument 1: Managing forests and woodlands for wood fuel is crucial: If forest conservation and management fail to save forests, biomass energy sources will shrink. When communities do not have alternative, affordable and accessible energy sources, they will go to great lengths to get wood for cooking and lighting. Hence, a failure in the system in one location can have subsequent adverse effects in surrounding areas [21,22].

Argument 2: Managing forests and woodlands for hydrological services is critical. Forests and woody vegetation in general play a crucial role in the hydrological processes in a landscape. With forest destruction (deforestation or forest degradation), sedimentation (or siltation) becomes a significant challenge for dams and reservoirs. When dams do not hold enough water for hydropower plants, power generation becomes limited. Then those people who rely on electricity will divert to biomass resources which then increases the pressure on forests and woody vegetation [23].

Argument 3: The high dependency on biomass energy is creating a vicious cycle of ecosystem degradation. With the current and projected high dependency on biomass energy in the next decades in Africa, unless forests and woodlands are conserved, restored, and managed, supplying energy for

the wider population will remain a critical challenge. If the current state of technological advances is not improved, it will further lead to a vicious cycle of ecosystem degradation, thus, exacerbating the problem [24–26].

Argument 4: Proper land use is fundamental [27]. Three dimensions can be addressed here.

1.  If land use in the watershed is not adequately regulated, soil erosion from upland areas leads to increased sediment accumulation in dams hence leading to Argument 2 [28].
2.  With poor land management, land productivity declines and consequently biomass for energy production, especially from energy crops and agricultural residues declines. This directly affects energy supply [29].
3.  With poor land management, livestock have limited feed sources and hence reduced biogas production potential.

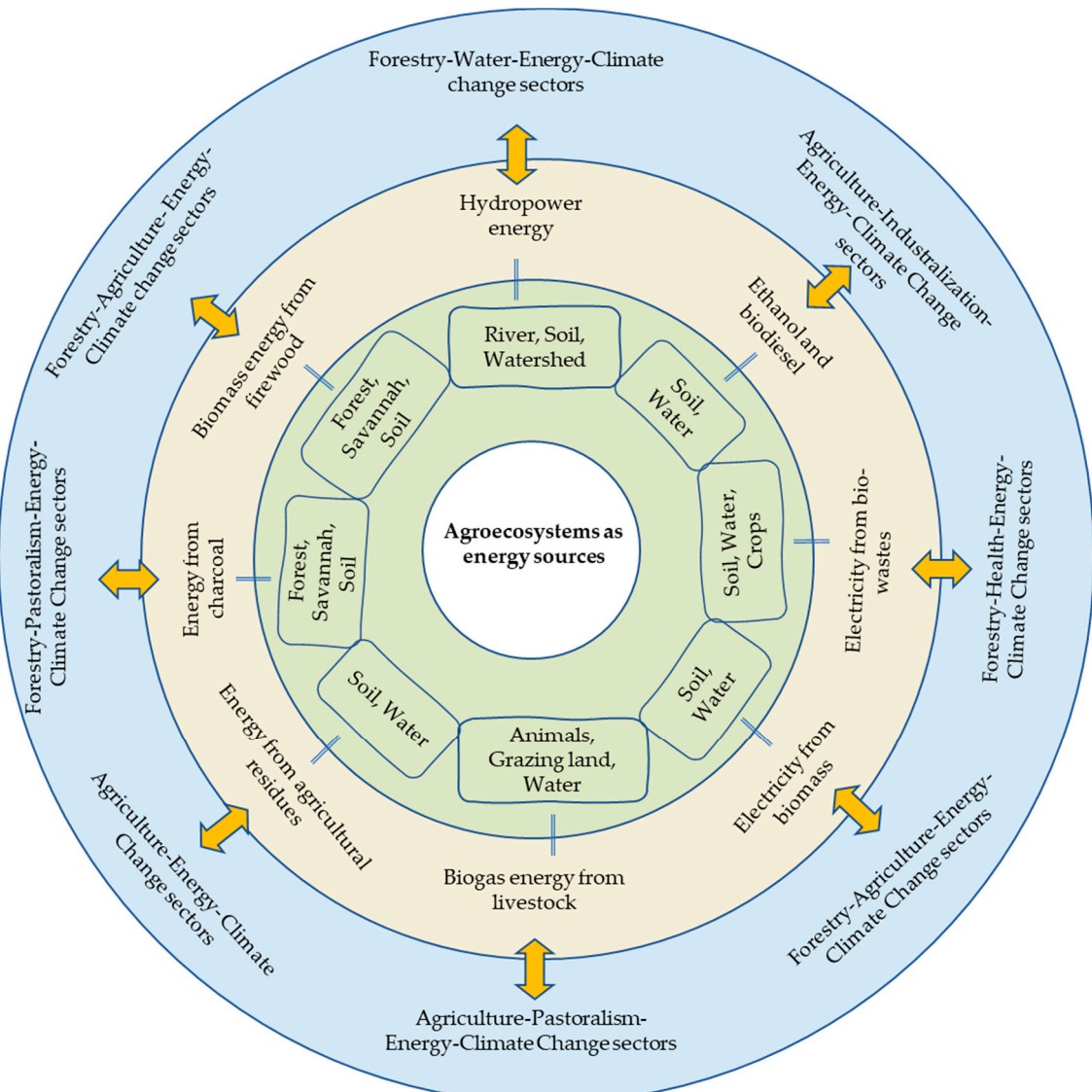

**Figure 1.** Authors' conceptual representation of ecosystem-based energy sources and their relations with different ecosystem elements and sectoral dimensions. Rectangular boxes indicate an immediate ecosystem element associated with the energy production process. The outer part of the bigger circle shows the interconnected sectors that strongly link to the energy production process.

Argument 5: Soil management plays a crucial role. If soil is not properly managed in watersheds, incoming moisture through rainfall often flows on the surface, joining river systems. This leads to reduced water infiltration into the soil and therefore, weak groundwater recharge. Silt-loaded rivers in turn diminish power generation [30].

Argument 6: Woody vegetation alone may not solve the ongoing energy scarcity on the continent [31]. Africa's population is growing, causing rising energy demand. If woody resources depletion is not abated, forests, savannah, and woodlands alone may not be able to sustain the supply of energy for the biomass dependent continent. Thus, there is a critical need for alternative sources of energy, proactive interventions that increase woody biomass, and technologies that enhance the conversion efficiency of energy raw materials. Residues from crops allow energy generation in forms such as biodiesel, ethanol, biowaste electricity and biogas energy to supplement the available resources [32,33] and reduce the pressure on ecosystems.

The prosperity of our ecosystems depends on how they are managed. If the different sectors indicated to have a strong linkage with energy (Figure 1) are not part of the whole planning process of energy supply strategies, the trade-offs among the sectors will increase, as observed across the tropics where agriculture remains the main cause of deforestation. It is important to note that no single sector stands alone. They are all linked, and part of the whole, the sustained management of which whose sustained management needs to embrace regenerative energy supply. Figure 2, building on the principles described in [17,19], shows pathways to a regenerative supply schema that could be widely beneficial. Due to limitations in space, not all pieces of the schema are dealt with in detail.

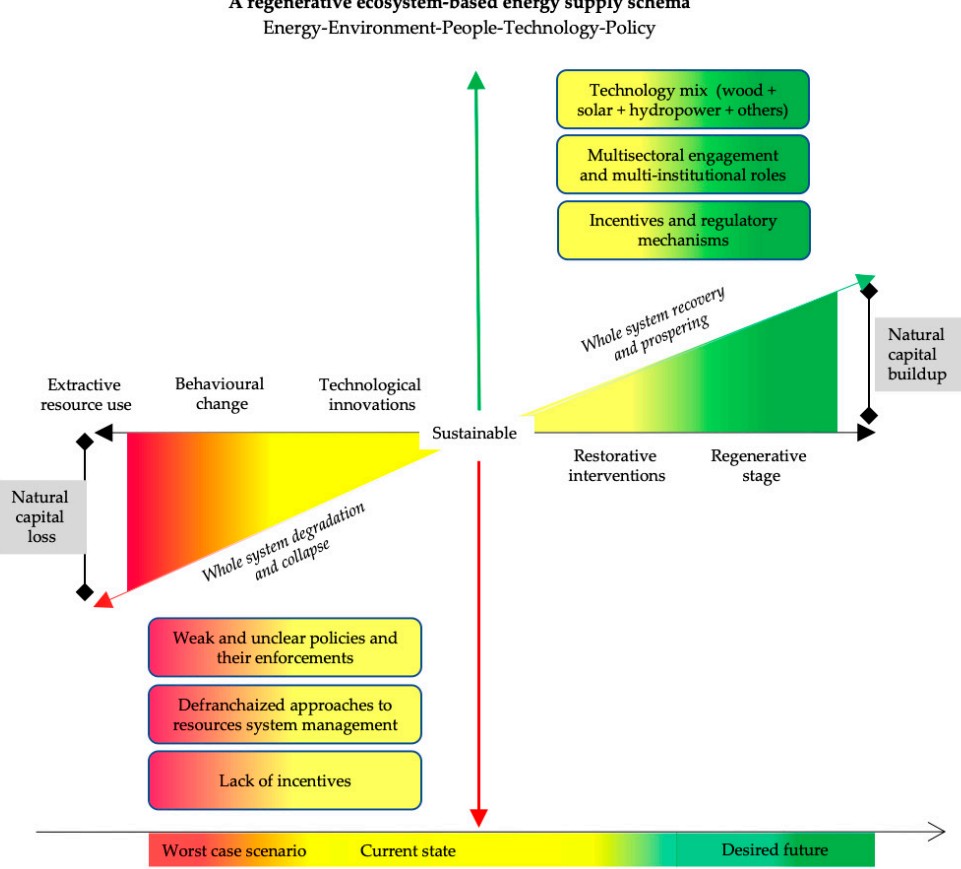

**Figure 2.** Redefining the path to regenerative ecosystem-based energy supply options based on the principles of a regenerative economy (adapted from Fullerton [17] and Reed [19]).

Reaching a desired future state of regenerative energy supply options requires a shift from the conventional way of thinking about and managing energy to a more holistic one. Sourcing energy must be considerate of the ripple effects it causes on the other components of agroecosystems as indicated in Figure 1. This can only be achieved if actions are taken to transition from the status quo (current modes of energy use) to a desired sustainable future condition (Figure 2). Several pathways could help.

1. Technologies to improve on efficiency: The energy production and use system in Africa is characterized by huge wastage of raw materials and very low raw material to energy conversion ratios. For instance, the most widely used cooking method on the continent, the three-stone fire, has an energy efficiency of less than 20% [34].

2. Technologies to transition to new forms of energy sources: Most of Africa's population still relies on firewood and charcoal for energy generation. However, with appropriate technologies, a shift to solar and wind energy, which are among the cheapest energy raw materials, can be possible. To date, the access rate of such technologies in Africa is exceedingly low.

3. Understanding the lifecycle analysis (footprints) of energy supply options: The energy that we use at every moment of our daily life comes at a significant environmental cost. For instance, generating one gigajoule (GJ) of energy from charcoal comes with a water footprint of about 53 m$^3$ [35]. Similarly, generating the same amount of energy from firewood (non-coniferous) comes with a water footprint of 21 m$^3$. Even further, the extraction of charcoal and firewood often causes deforestation and forest degradation, which subsequently threatens the habitat values of ecosystems and hence damages biodiversity.

## 3. Materials and Methods

### 3.1. The Scope of the Research

The current analysis focuses on Africa, particularly sub-Saharan Africa. In some instances, comparative examination was given to Latin America and Asia, both of which have similar sociodemographic and agroclimatic contexts—all three are tropical and subtropical regions. In the three regions, biomass energy and hydropower, to a large extent, dominate energy sources for households. The three regions are also the most affected by poverty and population growth, but particularly Africa and Asia. Hence, in some cases, a comparative insight looking at the three regions is presented although the major emphasis of the analysis is on sub-Saharan Africa.

### 3.2. Estimating the Ecosystem's Role in Supplying Energy

We estimated the contribution of an ecosystem to energy supply by focusing only on energy sources that are directly driven from land and water. In Africa, most studies have found that the biggest share of energy sources comes from land resources such as biomass. Hydropower is a growing energy supply means. Waste (both household and agricultural) was included in this assessment as it results from ecosystem management and affects the ecosystem too. In this study, we have excluded solar, wind, and geothermal power sources because these are not directly affected by the current and likely future contexts of ecosystem management. Though wind energy may be affected by the global circulation systems and seasonality, coherent and consistent data and evidence on this is scanty. Therefore, despite its relevance, we excluded it from the current analysis. For any specific period assessed, the energy supply potential of an ecosystem is the sum of the biomass tree-derived energy (firewood and charcoal), other biomass (e.g., stems of grasses, manure), hydropower, energy from waste, and biogas where data was available.

Data for biomass resources was obtained from a report on forest products from the United Nations' Food and Agriculture Organization [36]. Data for hydropower capacity was obtained from the International Hydropower Association [37]. Data for energy supply and production in energy units at country level for biofuels, waste, charcoal, and hydropower was collated from the United Nations Statistics Division [38]. Data availability for long-term analysis was a challenge, as available

data is limited to short time frames. For the data we used, we advise readers to consult the sources provided for any assumptions, caveats, and other considerations as to how the data was produced.

### 3.3. Estimating the Potential of Landscape Restoration to Boost Energy Supply in Africa

To estimate the potential of restoration for boosting energy supply, we collated data on the sparsely vegetated area (areas with low vegetation density) for each country in Africa from FAOSTAT [39]. We extracted data on areas of barren land, which currently have no vegetation cover. For this, we assumed half of it could be restored to grow woody biomass, which in turn affects the hydrological processes affecting the water flow into the river systems connected to hydropower reservoirs or dams. We assumed that there is a potential to restore all sparsely vegetated areas and that barren lands could be restored to produce biomass to the optimum capacity, which may vary by local agroecological contexts. This is not without recognizing the fact that some of such lands could be severely degraded and may require significant investments. Here, we use restoration scenarios with different restoration options having different biomass stocking potentials (Table 1).

**Table 1.** Various restoration options and their potential for emission removal and biomass production.

| Restoration Options | Annualized Emission Removal Rate (20 Years Range) (tCO$_2$ ha$^{-1}$ Year$^{-1}$) [40] | | Derived Biomass Equivalents (Tonnes a$^{-1}$ Year$^{-1}$) | |
|---|---|---|---|---|
| | Low | High | Low | High |
| Option 1: Natural regeneration | 9.1 | 18.8 | 5.28 | 10.91 |
| Option 2: Agroforestry | 10.8 | 15.6 | 6.27 | 9.05 |
| Option 3: Plantation—Broad-leaved species (tropical dry) | 10.1 | 11.3 | 5.86 | 6.56 |
| Option 4: Plantation—Broad-leaved species (tropical humid) | 21.4 | 29.2 | 12.42 | 16.94 |
| Option 5: Plantation—conifers (tropical dry) | 36.2 | 41.2 | 21.01 | 23.91 |
| Option 6: Plantation—conifers (tropical humid) | 20.8 | 26.4 | 12.07 | 15.32 |

We derived dry biomass equivalents for the emission values presented in Bernal et al. [40]. For conversions, we used an average wood fraction of 0.47 from the total biomass [41]. Potential biomass harvestable for energy from restoration options was computed using the area data for each country. The biomass was then converted into energy equivalents to estimate the potential. As this analysis is largely at the national level, final energy units generated may differ slightly as different countries may adopt a context-specific approach of restoring degraded vegetated areas and barren lands probably due to variations in agroclimatic zones resulting in different biomass production.

### 3.4. Estimating Energy Potential from Sparsely Natural Vegetated Areas

To estimate the potential of restoration for boosting energy supply, we collated data on sparsely naturally vegetated areas (areas with low vegetation density) for each country within Sub-Saharan Africa from [42]. For this, we assumed that restoration of half of these areas could grow a considerable amount of woody biomass, which in turn would affect the hydrological processes affecting the water flow into the river systems connected to hydropower reservoirs or dams. It is also important to note our assumption that all degraded vegetated areas may not be applicable or at least may require massive investment to restore the land. We assumed that all sparsely vegetated areas could be restored to accommodate biomass to the optimum capacity, which may vary according to the local agroecological and agroclimatic contexts.

The total biomass that can be utilized for energy is computed using the average annual biomass production of 4.3 tonnes/ha/year and half the sparsely naturally vegetated area in the country. The average annual biomass production is considered an average figure for all the countries due to the variety in the state of regions from total degradation to sparsely vegetated areas [43]. With the amount of biomass produced, the energy that can potentially be generated is computed using 20 GJ/tonne of energy production from dry biomass [44]. To analyze the potential number of people that can be adequately supplied with energy from this source, we used the per capita annual consumption in Sub-Saharan Africa of about 28.76 GJ.

### 3.5. Estimation of the Potential of Crop Residues to Boost Energy Supply in Africa

To estimate the energy potential of crop residues, we assembled data from [39] on the production quantity of various crops within Africa. The top 19 crops with an annual average production quantity above 1,000,000 tonnes were chosen in the years 2013–2018. The annual average production quantity is then used to compute the average residue amounts for each crop with the crop to residue ratio (CRR). We collated CRR values from various literature sources. Once the residue amounts were obtained, we used these values to compute the potential energy that can be generated from these crops annually with the energy content of each crop. The energy content of the crops was also derived from various literature sources as presented in the later sections of this paper (Table 4). We assume that all crop residues can generate energy without any further processing. For each crop, the potential energy generated is used to compute the number of people it can adequately supply with energy with the average annual per capita consumption of 28.76 GJ.

## 4. Results

### 4.1. Ecosystems as Prime Sources of Bioenergy in Africa

Ecosystems provide a critical mass of energy that is relied on by the population of countries. In Africa, of energy consumed by households in 2016, 86.87% was derived from ecosystem-based energy sources, namely, biofuels, waste, and electricity excluding charcoal due to lack of reliable data as it largely is sourced illegally. In absolute terms, this is about 10.5 million terajoules (2917 billion kilowatt-hours (kWh)) in 2016 alone. This translates to 2399 kWh per capita per year. In general, 35 out of 54 countries we assessed derived more than 75% of the household level consumption from ecosystem-based energy sources (Figure 3).

In the subsequent sections, we delve deeper into the different ecosystem-based energy sources, the future of which entirely depends on how well ecosystems are managed.

#### 4.1.1. Firewood

Firewood is the cheapest source of energy available that most people use widely. Consisting mostly of fallen sticks or branches, prunings of living or dead branches removed from standing trees, and wood from cut or felled trees, it is sourced from forests, woodlands, shrublands and in some cases from trees on farms (scattered trees, agroforestry, or energy woodlots). According to FAO [36], between 2013 and 2017, the total volume of wood fuel produced globally was about 9.44 billion $m^3$ with an average annual production of 1.88 billion $m^3$ [45]. Three-quarters of global wood fuel production and consumption is in Africa (35%) and Asia (39%). The tropics and subtropics (i.e., Africa, Latin America, and Asia) hold 88.3% of the global share of wood fuel production. In many developing countries, it is the most dominant source of energy (e.g., in Ethiopia—80% of the population [45] and over 90% in rural areas [2], Kenya—75% of the total energy supply [46]). However, as of 2017, people in Sub-Saharan Africa consumed more than twice the global average of wood fuel: 0.54 $m^3$ per person per year against 0.25 $m^3$ of wood fuel per person per year globally [36].

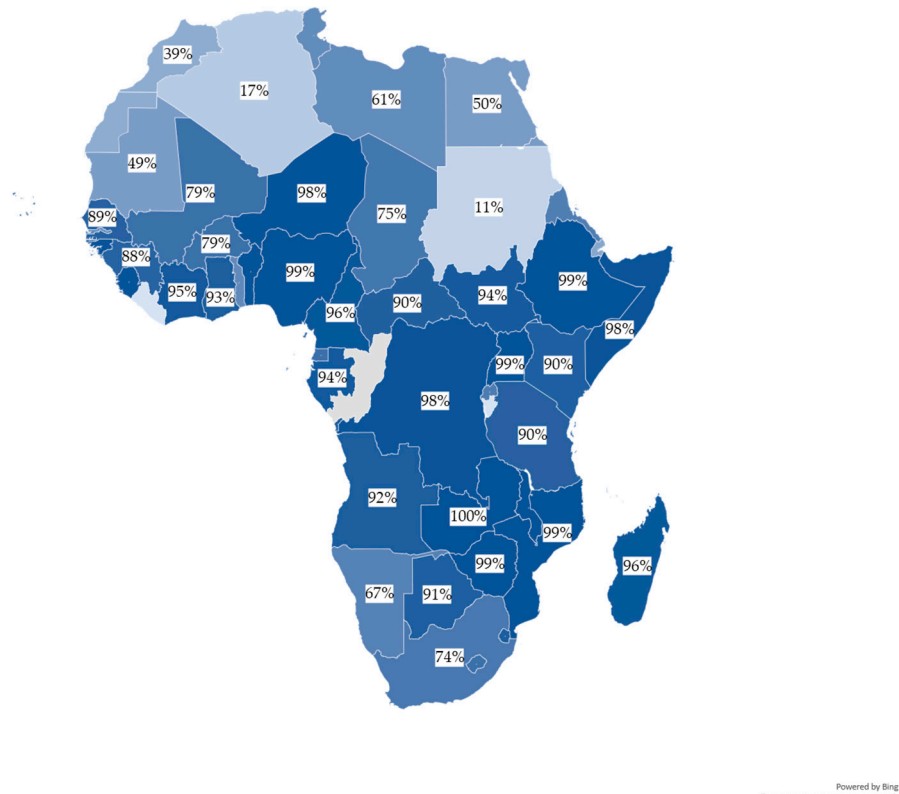

**Figure 3.** Share of energy consumed by households generated from ecosystem-based sources in 2016. Source: Data collated from [38] and analyzed using Microsoft Excel.

Duguma et al. [2] highlighted the impact of litter raking for firewood on soil quality in wooded areas. Under normal circumstances, litter and deadwood [47] decompose and enrich soils in forests and woodlands. However, as firewood collection has increased due to population increase, firewood extraction from vegetated areas has increased, leading to poor soil conditions and poor state of regeneration for the dominant tree species.

The intensive removal of firewood additionally diminishes the volume of organic material available for the active soil microbiota, thereby affecting soil aeration and water infiltration. Though firewood collection often focuses on dead wood and dry branches that fall off the trees and shrubs, when scarcity occurs, people begin to cut mature trees and split the wood to dry for firewood. Studies from Jimma, Ethiopia [48] indicate that the harvest rate of wood fuel is almost three times the annual permissible harvest in the forest and among trees outside forests. This subsequently leads to habitat degradation, loss of biodiversity, and losses or degradation of ecosystem services provided by the forests.

Woodfuel sourcing also affects water resources, especially in dry and semi-dry ecosystems and thus subsequently has an impact on hydropower generation. To comprehensively understand the effect of wood sourcing on the water source, we computed the total water footprint caused by wood fuel production in Africa. The average annual wood fuel production in Africa, between 2013 and 2017, was estimated at 1.41 billion $m^3$. Using the conversion factor of 0.09 $m^3$ roundwood/GJ [35], the average annual production of wood to energy equals 15.67 billion GJ. With the water footprint of firewood for non-coniferous wood at 21 $m^3$/GJ [35], Africa's water footprint from firewood consumption is estimated at 329 million $m^3$.

### 4.1.2. Charcoal Wood

About two-thirds (62%) of the global charcoal consumption between 2013 and 2017 was in Africa. Per capita, charcoal consumption in Africa was also the highest with 26 kg per person per year. This is 3.5 times the global average which stands at 7 kg per person per year [36]. At the country level,

45 countries in Africa had per capita consumption greater than the world average, and 37 countries had consumption rates of greater than three times the global average in 2017.

People who produce charcoal wood often focus on tree species providing high charcoal quality. Hence, there is significant species level degradation due to such targeted exploitation/utilization. These trees are often important sources of fruit and other edible tree parts for the population (e.g., the shea tree) and also serve as habitat and feed sources for a multitude of wild and domestic animals (e.g., insects, birds, bats, livestock, and others). The wood wastage associated with charcoal production scheme significantly affects how many trees charcoal producers should cut. Hence, in as much as charcoal production affects the ecosystem in many ways, ecosystem degradation also affects the future of charcoal production as the feedstock becomes scarce. Additionally, charcoal production techniques used in carbonization leads to wastage due to the low efficiency ranges. This is the case for the traditional earth kilns commonly used whose efficiency is 8–20% [49]. The growing awareness of the environmental consequences of charcoal is now leading to the ban of charcoal in many countries, for example, Gambia, Kenya, and other countries where vegetation recovery rate is slow. Nonetheless, despite the ban, charcoal markets and the utilization thereof have continued illegally. This has been exacerbated by the lack of alternative energy sources in the growing urban population in the continent.

The extent of consumption of charcoal sheds light into the amount of wood consumed within the continent for energy purposes and ultimately, the water footprint of charcoal. The average annual charcoal consumption in Africa, in 2017, was estimated at 31.5 million tonnes. Using the higher heating value of hardwood at 20.0 GJ/tonne [50], the average annual consumption of charcoal equals 630.76 million GJ. With the water footprint, based on charcoal consumption at 59 $m^3$/GJ [35], Africa's equivalent water footprint from charcoal consumed in 2017 is estimated at 37.21 billion $m^3$.

### 4.1.3. Hydropower

As of 2017, IEA estimated that, globally, there is a hydropower installed capacity of around 1267 GW, generating close to 4185 terawatt-hours of clean electricity; equivalent to about two-thirds of all renewable electricity generation [51]. Africa had the least share of this power source at 3% of the global share (Figure 4).

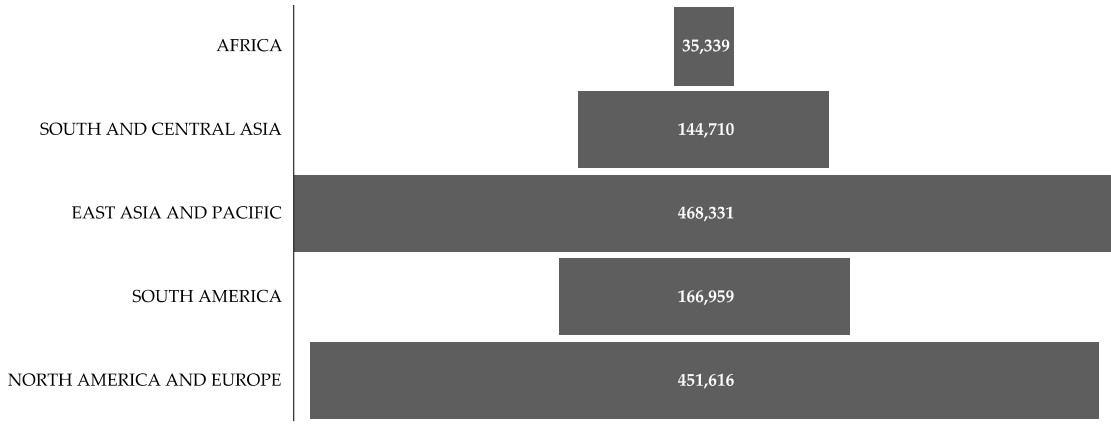

**Figure 4.** Total hydropower installed capacity (MW) in the different parts of the world as of 2017. Data source: [52].

Hydropower generation is one of the most ecosystem dependent energy production schemes. It significantly relies on the volume of water available, and any factor that affects water volumes significantly influences hydropower generation. Hydropower plants face significant challenges when the surrounding ecosystems are not managed sustainably, such as siltation in water reservoirs due to unsustainable farming systems, loss of water from evaporation due to degradation, and deforestation of supplying watersheds. In a study conducted in Songkhram River Basin, Thailand, Shrestha et al. [53]

estimated that the combination of land use change and climate change might lead to a decreased future streamflow of about 16%; significantly reducing the power generation capacity of the river system.

When hydropower generation is limited due to changes in water volume, the need for alternative affordable cooking and lighting energy increases. The most affordable and readily available sources of such alternative energy are firewood and charcoal [54]. However, it is essential to note that the tropics and subtropics are straining to supply these cheaper energy alternatives.

Similarly, various studies have highlighted negative consequences associated with hydropower generation, as shown in Table 2.

**Table 2.** Environmental consequences of hydropower generation schemes.

| Environmental Consequences | Description of Consequences or Impacts |
|---|---|
| Water | Disrupting the natural ecology of rivers by the plant establishment [55]. Deteriorating water quality—untreated domestic waste and runoff from agricultural or industrial uses. Dams and reservoirs change natural water temperatures and water chemistry [56]. Siltation—due to intense rainfall with erodible soils. |
| Soil | Eutrophication—the release of cyanotoxins since warm water conditions may accelerate the dynamics of the eutrophication process [57]. Soil erosion as runoff water carries away topsoil that is critical to produce food, feed, and fiber. |
| Atmosphere | The high amount of greenhouse gases released—industrialization leads to GHG emissions as well as other air pollutants. |
| Biodiversity | Damaging biodiversity—Hydropower dams can destroy habitats of aquatic life as well as inhibit the migration of fish [58]. |
| Landscape | Land use change—deforestation and clearing of land to set up hydropower plants, especially for large scale plants where dams need to be built. |

*4.2. Estimating Potentials of Ecosystem-Based Approaches to Bioenergy*

4.2.1. Ecosystem Restoration with Bioenergy Co-Benefits

Four different types of restoration are being looked at—degraded forest restoration, sparsely vegetated areas rehabilitation, grassland restoration, and trees on farms. The following sections detail the potential of each option.

Restoring degraded forests and savannah holds the greatest potential to generate bioenergy. Vast areas of Africa have experienced the recent clearing or degradation of its vegetation. Cai et al. [59] found that the total forest land area that currently has positive marginal productivity is about 196 million ha. When the savannah and shrubland areas are included, the figure increases by 323 million ha. In general, the estimated area of woody vegetation that has the potential to be restored is close to 529 million ha.

Considering the challenges in implementing restoration schemes, namely, limitations in inputs, technical capacity, and other readiness attributes, we assume that only half of the land area available for restoration can be restored soon. This makes the active intervention area 264 million ha. Due to variabilities in the state of degradation, in other words, ranging from areas that are sparsely vegetated to those experiencing total degradation, we use a conservative average aboveground biomass increment of 4.3 tonne/ha/y [43]. It gives an average annual biomass production of close to 1.14 billion tonnes. Using the 20 kJ/gm energy production of dry biomass [44], the aggregate energy produced is equivalent to $22.8 \times 10^9$ GJ. With the per capita annual consumption of SSA of about 687 kg oil equivalent (7990 kWh or 28.76 GJ) (as of 2014), the energy produced from restoration could suffice for the annual energy needs for 0.79 billion people at the current level of energy use with complete biomass to energy conversion. Although most of the African population uses technologies that have biomass to energy conversion efficiencies of not more than 40%, the potential energy estimated above could suffice for

about 316 million people. This is equivalent to the combined populations of Nigeria and Ethiopia, the two most populous nations in the continent, combined as of 2019.

Such potential could be realized by deploying techniques such as enrichment planting, reforestation, and assisted natural regeneration. These would help in restocking biomass in degraded areas without changing the land use type from forests. The sustainability aspect of this approach is that the wood fuel harvest would not involve harvesting the entire biomass stock but rather the annual biomass increments alone.

Restoration of sparsely vegetated areas holds an immense potential for generating bioenergy through restoring ecosystems. According to FAOSTAT, Africa, as of 2015, had close to 80,785,000 ha of land covered with sparsely vegetated areas—areas with vegetation cover between 2 and 10%. Restoring this area will considerably abate the energy scarcity that the continent's population is facing and is likely to continue to face if ecosystems are not restored. Considering the logistical and technical challenges, we again assume that only half the area available is used for restoration. This makes a total of about 40,392,000 ha of land with the potential to generate 173,687,750 tonnes at about 4.3 tonnes $ha^{-1}$ $y^{-1}$ [43]. This could produce about 3.474 billion GJ at 20 GJ/tonne of energy content. At a consumption rate of 28.76 GJ per capita per year, the energy generated would suffice for about 121 million people.

Grassland area rehabilitation is another opportunity for bioenergy generation through ecosystem restoration. Field et al. [44] estimated that tropical grasslands have the potential to produce aboveground biomass of 7–20 tonnes $ha^{-1}$ $y^{-1}$ with a net aboveground biomass growth of 4.3 tonnes $ha^{-1}$ $y^{-1}$. Cai et al. [60] estimated that abandoned grasslands in Africa occupy close to 26 million ha. We again considered restoring only half the area of the abandoned grasslands, also noting that grasslands are important ecosystems in their own right. If these were to be implemented, however, it could generate close to $1.118 \times 10^9$ GJ of energy. Using the per capita annual consumption of 28.76 GJ, the biomass energy produced from restored grasslands could suffice for the energy needs of 41 million people on the continent at full potential. At 40% biomass to energy conversion efficiency (depending on the most efficient stove available to the society), 16.4 million people could be adequately supplied by the resulting energy.

### 4.2.2. Trees on Farm (Agroforestry) Promotion

Cai et al. [59] estimated that Africa has close to 133 million ha of cropland (65 million ha of cropland and 66 million ha of cropland-vegetation mixed) with marginal productivity. Campbell et al. [43] estimated the weighted mean biomass production in such lands to be about 4.3 tonnes $ha^{-1}$ $y^{-1}$, the biomass produced could be equivalent to 0.6 billion tonnes $y^{-1}$ with a potential to generate 11.44 billion GJ. Assuming only 40% of this becomes effective energy that can be used for heating and cooking by SSA population, it may be sufficient for 159 million people—almost equivalent to the combined population of Tanzania, Uganda, and Kenya as of 2019 and a vast potential that the continent could capitalize on.

Suppose due to technical and other capacity challenges only 60% of the available land area is converted to biomass growth. In that case, the energy produced could fulfill energy needs of about 95.4 million people, slightly less than the combined population of Kenya and Uganda as of 2019. The production of biomass for bioenergy on such degraded agricultural lands could be done using fast-growing adaptable species. Table 3 gives details of some high energy value tree species.

**Table 3.** Energy attributes of different tree species that can be grown for energy in Africa.

| Species | Growing Zones | Mean Annual Increment (m³) [a] | Calorific Value of Wood (KJ/gm) | Means of Propagation |
|---|---|---|---|---|
| *Gliricidia sepium* [b] | Tropics | (2.84–3.19) [1–3] | 4.5 | Cuttings |
| *Acacia mearnsii* [c] | Tropics and subtropics | 10–25 [7–10] | 3.7 | Seed |
| *Acacia nilotica* [d] | Tropics | 3–5 [15–20] | 4.9 | Seedlings |
| *Acacia polyacantha* | Wooded grassland, deciduous woodland | N.D. [15] | 4.0 | Seeds |
| *Acacia xanthophloea* | Tropical and subtropical savannah | 1.0–1.5 [20–30] | 4.4 | Seeds |
| *Terminalia brownii* | Arid, semi-arid, and sub-humid | N.D. | 4.6 | Seeds |
| *Terminalia orbicularis* [e] | Arid regions | N.D. | 5.1 | Seeds |
| *Commiphora africana* | Dry savannahs | N.D. | 4.8 | Cuttings |
| *Commiphora baluensis* [f] | Arid, semi-arid rangelands | N.D. [10–20] | 4.4 | Seeds |
| *Grevillea robusta* [g] | Riverine rainforest | 5–15 [8–10] | 4800 (kcal/kg) | Seeds and cuttings |
| *Markhamia lutea* [h] | Lake basins and highland areas | 20 | N.D. | Seeds |

[a] It represents the mean annual increment of the aboveground biomass and numbers in parenthesis denote rotation periods.; [b] [60]; [c] [61]; [d] [62]; [e] [62]; [f] [63]; [g] [64]; [h] [65]; N.D denotes No Data.

Among the most feasible ways to produce biomass energy are private energy woodlots, on-farm tree growing, and communal woodlots. It should also be noted that depending on their planting preferences, farmers and pastoralists could grow trees on their own land that could be sufficient for their annual energy needs. In Kenya, Njenga et al. [66] found that if farmers could grow 176 trees in any arrangement on their farm plots, pruning from the trees is sufficient to provide all fuelwood needed. Iiyama et al. [49] showed that if the rising charcoal demand in Africa is ignored, the threat to forest ecosystems is going to be intense. The authors propose agroforestry as an alternative solution to complement the wood fuel supply.

### 4.2.3. Agricultural Residue and Waste for Bioenergy Generation

Africa is highly dependent on agriculture and many crops are produced extensively on the continent. To compute the energy potentials of the crop residues, we selected the 19 most widely produced crops. We used the crop to residue conversion ratio (CRR) (see Table A1). The results showed that residues could generate the energy equivalent to 10.52 billion GJ, sufficient to support close to 366 million people (around 33% of the sub-Saharan population as of 2019) annually at 28.76 GJ per capita consumption if the residue is prepared and packaged ready for energy generation (Table 4). However, in some countries, residues directly used for cooking and heating depend on the availability of technologies to convert the residue to energy. With a mean biomass energy potential of 20 GJ/tonne, the amount of woody biomass saved is 526 million tonnes of wood which is equivalent to the mean annual increment from 122 million ha of forested areas at a mean annual increment of 4.3 tonnes ha$^{-1}$ y$^{-1}$.

The use of crop residues for energy generation has often generated controversy among agricultural experts as it leads to exporting of nutrients from the farm [32,67]. However, in Africa, even if residues remain on the farm before the rainy season begins, farmers burn the fields to get rid of pests and obstructions from the previous year's farming. Hence, residues from traditional agricultural practices do exist and could be used to generate energy. Furthermore, the ash produced could be used to fertilize fields. Thus, generating energy from crop residues could be instrumental if the enablers for its use are put in place. This needs investment from the relevant sectors but will have the additional benefit of reducing the pressure on forests and trees. Final values differ slightly according to each crop due to the difference in production quantities, CRR, and energy content values.

Yam, the most widely produced food crop in Africa, has an average annual production of about 67 million tonnes. It is excluded from the analysis as data on the energy content and CRR values were not obtainable.

**Table 4.** The energy generation potential from crop residues in Africa.

| Crop Type | Average Annual Production Quantity (2013–2018) (Tonnes) [a] | Energy Content (GJ/Tonne) [b] | Total Energy Potentially Generated (Million GJ) | Number of People That Could be Supported (In Million at 28.76 GJ/Year Per Person) |
|---|---|---|---|---|
| Maize | 76,864,066.67 | 12.46 | 2394.32 | 83.25 |
| Wheat | 26,988,158.50 | 17.00 | 1628.74 | 56.63 |
| Rice (paddy) | 31,443,454.67 | 13.45 | 1353.33 | 47.06 |
| Groundnuts | 13,405,968.50 | 18.80 | 1279.57 | 44.49 |
| Millet | 13,068,719.17 | 20.00 | 771.05 | 26.81 |
| Sorghum | 28,046,571.83 | 12.38 | 434.02 | 15.09 |
| Sugar cane | 93,947,959.83 | 16.60 | 389.88 | 13.56 |
| Potatoes | 25,468,619.33 | 17.50 | 361.02 | 12.55 |
| Barley | 6,747,087.83 | 17.00 | 349.84 | 12.16 |
| Coffee | 1,137,476.00 | 12.69 | 303.13 | 10.54 |
| Cow peas | 6,631,827.17 | 15.00 | 288.48 | 10.03 |
| Seed cotton | 4,467,497.50 | 17.65 | 279.92 | 9.73 |
| Soybeans | 2,748,016.17 | 15.00 | 232.48 | 8.08 |
| Cassava | 168,741,313.67 | 17.50 | 183.08 | 6.37 |
| Oil palm fruit | 18,957,246.17 | 13.26 | 106.83 | 3.71 |
| Coconuts | 2,050,956.17 | 17.40 | 61.38 | 2.13 |
| Beans | 6,572,401.50 | 16.00 | 52.58 | 1.83 |
| Sweet potatoes | 25,311,996.00 | 17.50 | 44.3 | 1.54 |
| Onions | 11,391,041.67 | 10.49 | 5.97 | 0.21 |
| | | Aggregate | 10,519.92 | 365.78 |

[a] Data source: [39]. [b] Data source: See Table A1 for details.

## 5. Discussions

### 5.1. Policy Mechanisms for Ecosystem-Based Approaches to Bioenergy

Several policy initiatives aimed at promoting ecosystem-based bioenergy have been formulated to tap the existing African bioenergy resources and potentials. At the continental level, the Africa Bioenergy Policy Framework and Guidelines [68] give principles for individual African countries to develop environmentally friendly and socially responsible bioenergy policies and regulations. They outline options to remove financial barriers, develop incentives, and reward bioenergy expansion. The aim is to unlock existing potentials, establish effective bioenergy policy monitoring systems, raise awareness of bioenergy resources, as well as develop public-private partnership models to exploit bioenergy resources. At the national level, some countries have developed national monitoring strategies for harnessing bioenergy potential to position themselves for bioenergy competitiveness and address associated risks (Table 5).

**Table 5.** Examples of bioenergy related policies and strategies in selected African countries.

| Country | National Bioenergy Objective/Mission | Relevant Policies/Strategies |
|---|---|---|
| Ghana | Modernize and maximize the benefits of bioenergy on a sustainable basis. | Bioenergy Policy for Ghana, 2010 [69] |
| Kenya | Facilitating provision of clean, sustainable, affordable, competitive, reliable, and secure energy services at the least cost while protecting the environment | National Energy and Petroleum Policy, 2015 [70] |
| Rwanda | Switching from traditional to modern biomass energy sources that have socio-economic, health, and environment benefits. | Biomass Energy Strategy 2019–2030 [71] |
| Mozambique | Operationalizing the biofuel policy and strategy | Biofuel Sustainability Framework, 2014 [72] |
| South Africa | Stimulating rural development and reducing poverty by creating sustainable income-earning opportunities sector | Biofuels Industrial Strategy, 2007 [73] |

These policies recommend significant incentives towards sustainable harnessing of bioenergy potential. Incentives such as tax credits, public-private partnership programs, grants, and tax exemptions are employed to promote bioenergy products and consumption. Incentive mechanisms towards lands and forestry restoration such as reduced emissions from deforestation and forest degradation (REDD+) have the potential to promote bioenergy stocks, reduce biodiversity loss, and cut

down emissions [74]. REDD+ can also promote bioenergy development by incentivizing stakeholders to increase tree cover to mitigate climate change. This offers double benefits in terms of increased biomass potential.

*5.2. Multisectoral Engagement and Multi-Institutional Roles*

Multisectoral engagement and harmonizing multi-institutional roles are enablers in developing effective bioenergy policies at different levels. Due to the interlinkages and consequently direct and indirect impacts, systems thinking is required within policy development. Different tiers of stakeholders are involved, directly or indirectly, in the development and execution of bioenergy-related policies. National and local governments are primarily involved in policy development and implementation. Government agencies are also essential players; bioenergy resources require regulation and monitoring. The commercial sector is made of up of bioenergy source producers, distributors, advisory agencies, and providers who work in tandem with government agencies in the process. Financial institutions such as banks and micro-finance institutions are also essential providers of business support, resources, and financial advisory services in bioenergy development. Bilateral and multilateral organizations, regional bodies, economic blocks, and development agencies play an essential role in setting global guidelines and standards that can be domesticated at a national level. Non-governmental organizations such as community-based organizations, trade organizations, and religious groups are influential as well at local levels. With such a range of stakeholders, a harmonized strategy is essential to engage all players in bioenergy development.

*5.3. Enforcement of Rules, Regulations and Provisions Relevant to Sustainable Bioenergy*

Enforcement of rules and regulations relevant to sustainable bioenergy interventions is primarily guided by existing national policies and domesticated international standards and guidelines. Critical elements of the national regulatory framework include policy development, institutional frameworks, market and trade regulations, creation of incentives, certification schemes, and establishing precise research and development mechanisms [75]. Efficient production systems must be consistent with national regulations and policies such as environmental standards, pollution control, and forestry and agricultural policies. Enforcement of bioenergy policies, rules, and regulations also require effective coordination among different sectors, parties, and institutions playing different roles along the bioenergy value chain [76]. In essence, the enforcement process should consider the landscape ecosystem functioning in terms of trade-offs and complementarities.

Rules and regulations facilitate the provisions for implementation of the right interventions to promote sustainable ecosystem-based bioenergy supply options. We looked at various scenarios (Figure 5) and their implications for bioenergy generation potentials. Our findings reveal that even if the four activities are implemented at 30% of the existing potential, the energy generated could be enough to fulfill 56% of the annual energy needs of the continent, assuming that the right technologies for converting the biomass to energy are employed. At 50% implementation, close to 94% of the population of Sub-Saharan Africa could have access to domestic energy supply. Even if we assume, due to legal aspects (e.g., tenure rights and protection), that forest areas may not be accessed for energy purposes, at 30% and 50% implementation rates, restoring of sparsely vegetated areas, the practice of agroforestry, and the use of crop residues could potentially generate bioenergy that is sufficient to fulfill 35% and 58% of domestic energy needs, respectively.

This result implies that Sub-Saharan Africa can sustainably support itself without causing further damage to its precious ecosystems by adopting interventions that boost ecosystem-based approaches to bioenergy supply (see Figure 5). If the domestic energy (at household level) could be supplied through such means, the supply from other energy sources such as solar, hydropower, geothermal, and others could be diverted for industrial purposes, thereby propelling economic growth. The continent, however, needs to invest in technologies needed for converting biomass into energy in an effective

manner and put in place resources needed to restore ecosystems to supply the envisaged energy and other benefits.

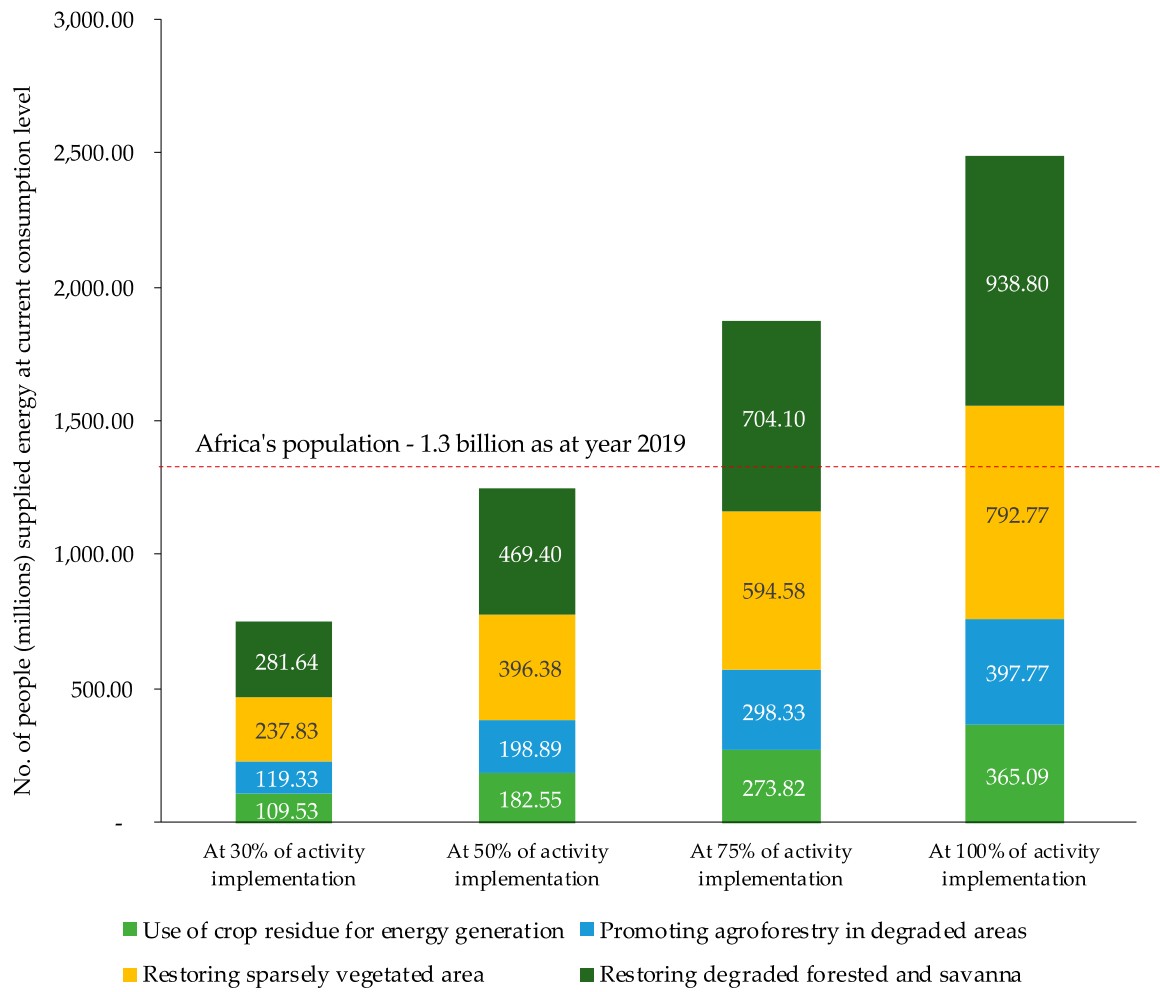

**Figure 5.** Estimation of the number of people that the different land-based practices could support under different implementation scenarios at per capita consumption rate of 28.76 GJ annually. The estimated energy values are assuming mean annual biomass increments for all forest, savannah, sparsely vegetated areas, and agroforestry.

*5.4. A SWOT Overview of Ecosystem-Based Bioenergy Interventions*

Ecosystem-based bioenergy interventions can make a great contribution to the sustainability of the fragile ecosystems that sustain us. They can reduce GHGs [44], improve soil structure and fertility, and reduce erosion. Bioenergy as a source is also more reliable than solar or wind energy due to the scant energy loss while in operation. Bioenergy systems conserve natural resources by reducing the use of non-renewable resources, maintaining the quality of resources, or recycling and reusing resources.

Low adoption of ecosystem-based bioenergy interventions is associated with inadequate enabling national policies and regulatory frameworks needed to attract commercially viable investment, especially in Africa [77]. Despite its enormous potentials, the big question is often—who would be willing to invest in such schemes? Technical and technological weaknesses, including trained personnel, necessary equipment, and infrastructure, are a major hindrance as Luthra et al. [78] notes. Most of these technologies are not locally available, thus attracting high importation, installation and maintenance costs and consequently low adoption. There is also a general lack of incentives such as credit facilities and subsidies in many African countries to enhance more investments in these interventions despite their huge potential to address energy demands. From the marketability perspective, some of the

key barriers are high costs of bioenergy production and service delivery, competition from traditional non-renewable sources such as fossil fuels, poor market development of ecosystem-bioenergy sources, and the finite (non-renewable) nature of bioenergy raw materials [79], which have as a result reduced their adoption rates.

New employment opportunities arise from growing and harvesting biomass, transporting, handling, and through procurement, construction, operation, and maintenance of bioenergy plants could potentially create 1.5 million direct jobs globally [80]. Bioenergy establishment provides a platform for rural development through local actors' involvement in training and capacity building, participation in innovative financial mechanisms, and increased income through the sale of biofuels [81]. However, bioenergy production poses environmental risks like nutrient mining and land degradation, depletion of water resources, monoculture, methods of harvesting that expose land to more significant erosion, and pollution from pesticides and fertilizers [82]. Most biofuel crops are invasive or potentially invasive due to their traits such as rapid growth rate, low maintenance, and high yields. Some areas do not support the growth of bioenergy crops due to unsuitability of the land, degradation, the inadequacy of current policies, and resource quality difference in geographical areas.

## 6. Conclusions

Sub-Saharan Africa, despite its deteriorating state of ecosystems, is still highly dependent on biomass for energy generation for domestic consumption. Around 87% of the continent's energy supply is directly ecosystem dependent. In 2016 alone, the continent generated about 2917 billion kWh of energy from firewood, charcoal, and hydropower.

Nevertheless, with proper technical and policy support, ecosystem-based energy production holds great potential. The total energy generation potential from ecosystem-based energy sources is 27 billion GJ (restoring of degraded forest and savannah), 22.80 billion GJ (restoring of sparsely vegetated areas), 11.44 billion GJ (promotion of agroforestry in degraded farm areas), and 10.50 billion GJ (use of main crop residues for energy). This indicates the substantial potential restoration has for fulfilling the energy needs of the continent for domestic use at least. Investing in such interventions will not only secure energy for population but also sustainably enhance the supply of ecosystem services that are crucial for human survival. At an implementation level of 50% (i.e., if the continent commits to invest in 50% of the potential identified), Sub-Saharan Africa can supply energy to its population even if technologies with low biomass to energy conversion potentials are used. Restoring ecosystems while generating sufficient bioenergy has the potential to drive a transition from the current extractive mode to regenerative approach. Achieving this would reduce rates of forest and woodland losses from clearing and selective logging. The regenerative aspect of these energy supply options is also that at any point in time there is no full harvest of the main stock but rather the bioenergy supply depends on the mean annual increments of the restored ecosystems.

This study looked at continental scale of energy issues. It is undoubtedly necessary to contextualize the options proposed to suit local country or field level contexts to achieve the best outcome. It is also important to note that estimates of areas available for ecosystem-based bioenergy supply are broadly optimistic, relying largely on previous studies. Under local conditions, what may seem to be degraded pastureland may be providing critical ecosystem services that the local community may not want to forgo to increase woody biomass production for use as energy. In summary, the estimates are based on best available material but may not be so precise. Despite such assumptions and limitations, this study should ignite the discourse on why it is crucial to look at energy issues as a multisectoral issue and as a potential driver of a regenerative economy. The indicated pathways in the study provide national governments with the win-win scenarios to directly fulfill global targets on sustainable development goals (SDG) such as SDG 7 (affordable energy), SDG 13 (climate action) and SDG 15 (life on land).

**Author Contributions:** Conceptualization, L.D.; methodology, formal analysis, L.D. and E.K.; writing—original draft preparation, L.D., E.K., J.N., P.A.M. and K.M.; writing—review and editing, L.D., E.K., J.N., P.A.M. and K.M. All authors have read and agreed to the published version of the manuscript.

**Funding:** The authors acknowledge the Forest, Trees and Agroforestry (FTA) Program of the CGIAR for the financial support.

**Acknowledgments:** We are very grateful to the reviewers for the thoughtful and very useful inputs that helped improve the manuscript. The authors acknowledge the support the Forest, Trees and Agroforestry Program of the CGIAR for financial support. We are very grateful for the technical and editorial support from Claire Anulisa, Dibo Duba, Priscilla Wainaina, Himlal Baral, Cathy Watson and Yvonne Baraza.

**Conflicts of Interest:** The authors declare no conflict of interest.

## Appendix A

**Table A1.** Estimates of the energy contained in the commonly produced crop residues in Africa.

| Crop Type/Residues | Average Annual Production (Tonnes) (2013–2018) | Crop to Residue Ratio (CRR) | Residue Quantity (Tonnes) | Energy Content (GJ Per Tonne of Residue | Total Energy Potential (GJ) |
|---|---|---|---|---|---|
| **Cassava** | 168,741,314 | | | 17.50 [83] | |
| Stalk | | 0.062 [84] | 10,461,961 | | 183,084,32 |
| **Sugarcane** | 93,947,960 | | | 16.60 [83] | |
| Tops and leaves | | 0.05 [84] | 4,697,398 | | 77,976,807 |
| Bagasse | | 0.2 [84] | 18,789,592 | | 311,907,227 |
| **Maize** | 76,864,067 | | | 12.46 [83] | |
| Stalk | | 2 [84] | 153,728,133 | | 1,915,452,541 |
| Cob | | 0.3 [84] | 23,059,220 | | 287,317,881 |
| Husk | | 0.2 [84] | 15,372,813 | | 191,545,254 |
| **Rice (paddy)** | 31,443,455 | | | 13.45 [83] | |
| Stalk | | 1.5 [84] | 47,165,182 | | 634,371,698 |
| Husk | | 0.2 [84] | 6,288,691 | | 84,582,893 |
| Straw | | 1.5 [84] | 47,165,182 | | 634,371,698 |
| **Sorghum** | 28,046,572 | | | 112.38 [83] | |
| Straw | | 1.25 [84] | 35,058,215 | | 434,020,699 |
| **Wheat** | 26,988,159 | | | 17.00 [83] | |
| Stalk | | 1.5 [84] | 40,482,238 | | 688,198,042 |
| Straw | | 1.75 [84] | 47,229,277 | | 802,897,715 |
| Pod | | 0.3 [84] | 8,096,448 | | 137,639,608 |
| **Potatoes** | 25,468,619 | | | 17.50 [83] | |
| Stalk | | 0.05 [83] | 1,273,431 | | 22,285,042 |
| Tops and leaves | | 0.76 [83] | 19,356,151 | | 338,732,637 |
| **Sweet potatoes** | 25,311,996 | | | 17.50 [83] | |
| Stalk | | 0.1 [83] | 2,531,200 | | 44,295,993 |
| **Oil palm fruit** | 18,957,246 | | | 13.26 [85] | |
| Shell | | 0.065 [84] | 1,232,221 | | 16,339,250 |
| Fiber | | 0.13 [84] | 2,464,442 | | 32,678,501 |
| Brunches | | 0.23 [84] | 4,360,167 | | 57,815,809 |
| **Groundnuts** | 13,405,969 | | | 18.80 [85] | |
| Stalk | | 2 [84] | 26,811,937 | | 504,064,416 |
| Husk | | 0.0477 [84] | 6,394,647 | | 120,219,363 |
| Straw | | 2.3 [84] | 30,833,728 | | 579,674,078 |
| Shell | | 0.3 [84] | 4,021,791 | | 75,609,662 |
| **Millet** | 13,068,719 | | | 20.00 [86] | |
| Stalk | | 1.2 [84] | 15,682,463 | | 313,649,260 |
| Straw | | 1.75 [84] | 22,870,259 | | 457,05,171 |
| **Onions** | 11,391,042 | | | 10.49 [87] | |
| Stalk | | 0.05 [83] | 569,552 | | 5,974,601 |
| **Barley** | 6,747,088 | | | 17.00 [84] | |
| Stalk | | 1.3 [84] | 8,771,214 | | 149,110,641 |
| Straw | | 1.75 [84] | 11,807,404 | | 200,725,863 |
| **Cow peas** | 6,631,827 | 2.9 [84] | 19,232,299 | 15.00 [83] | 288,484,482 |
| **Beans** | 6,572,402 | | | 16.00 [83] | |
| Stalk | | 0.5 [85] | 3,286,201 | | 52,579,212 |
| **Cotton (seed)** | 4,467,498 | | | 17.65 [88] | |
| Stalk | | 2.9 [84] | 15,859,616 | | 279,922,225 |
| **Soybean** | 2,748,016 | | | 15.00 [89] | |
| Stalk | | 1.7 [84] | 4,671,627 | | 70,074,412 |
| Straw | | 3.94 [84] | 10,827,184 | | 162,407,755 |
| **Coconut** | 2,050,956 | | | 17.40 [83] | |
| Husk | | 1.6 [84] | 3,281,530 | | 57,098,620 |
| Shell | | 0.12 [84] | 246,115 | | 4,282,396 |
| **Coffee** | 1,137,476 | | | 12.69 [83] | |
| Husk | | 21 [84] | 23,886,996 | | 303,125,979 |
| Total | 563,990,378 | | 697,866,522 | | 10,519,921,758 |

Note: Numbers in parenthesis refer to the source of the data. Bold texts in the column 'crop type/residues' represent the common name of the agricultural crop.

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
