# Peer review of "Ecosystem-Based Approaches to Bioenergy and the Need for Regenerative Supply Options for Africa"

_sustainability, doi:10.3390/su12208588_

Round 1

Reviewer 1 Report

Reviewer.

In the article presented ecosystem-based approaches to bioenergy for Africa, and analyzed the need for regenerative supply options. This article has been prepared in sufficient quality, there are presented many and various analysis results, which were compared with the results of other researchers. Analysis results are sufficient original and new, but the novelty of the work in the article should be more highlighted. The manuscript is structured in agreement with instructions to authors, and the quality of presented paper is sufficient high, but inaccuracies and areas for correction were observed in the manuscript.  

Comments and remarks of reviewer:

  • All tables and the text are not fully prepared in accordance with the requirements. In the tables there is no need to draw vertical lines.
  • In the chapter “2 A conceptual framework: Towards regenerative energy supply options” provides a lot of information on various factors and activities, but this chapter is presented in an abstract way, it lacks specific data and results of analysis and research results (only at the end of this chapter presented some concrete data).
  • In Figure 1 presented a conceptual representation of ecosystem-based energy sources and their relations…, but it is not clear, is this information, presented in figure, is the work of authors or it is based on the information from other authors?
  • Too many small sections and subsections throughout the article, especially in the “4 Results” and “5 Discussion” chapters.
  • The graphs presented in Figure 5 are incorrect, it is better to present this information in separate columns.
  • It is not recommended to provide graphs or figures in the conclusions, so I suggest to move “Figure 8” to the Results chapter, with comments on the information in that figure as well.
  • The bibliographic descriptions of many information sources in the chapter “7 References” need to meet the requirements for authors.

Author Response

Reviewer comments and repsonses

In the article presented ecosystem-based approaches to bioenergy for Africa, and analyzed the need for regenerative supply options. This article has been prepared in sufficient quality, there are presented many and various analysis results, which were compared with the results of other researchers. Analysis results are sufficient original and new, but the novelty of the work in the article should be more highlighted. The manuscript is structured in agreement with instructions to authors, and the quality of presented paper is sufficient high, but inaccuracies and areas for correction were observed in the manuscript.  

Responses: We have made significant edits to the details of the text for clarity, consistency and correction of inaccuracies. We have annexed to this submission the track change file in addition so that the level of effort.

Comments and remarks of reviewer:

  • All tables and the text are not fully prepared in accordance with the requirements. In the tables there is no need to draw vertical lines. Response: All texts, figures and tables are now formatted using the template provided by the journal. We have made extensive work on this as can be visible from the supplementary file Track changed document.
  • In the chapter “2 A conceptual framework: Towards regenerative energy supply options” provides a lot of information on various factors and activities, but this chapter is presented in an abstract way, it lacks specific data and results of analysis and research results (only at the end of this chapter presented some concrete data). Response: This section is largely making the case for the regenerative energy supply options and on its own does not have any specific quantitative analysis to itself. That is why the results section is presented fully in section 4 following the methods part.
  • In Figure 1 presented a conceptual representation of ecosystem-based energy sources and their relations…, but it is not clear, is this information, presented in figure, is the work of authors or it is based on the information from other authors? Response: Figure 1 is the conceptualization of the broader framework of multisectoral nature of energy and the need for ecosystem-based approaches to bioenergy
  • Too many small sections and subsections throughout the article, especially in the “4 Results” and “5 Discussion” chapters. Response: The two sections are now substantively revised. Now much fewer sections than in the original version.
  • The graphs presented in Figure 5 are incorrect, it is better to present this information in separate columns. Response: We have decided to remove figure 5 as most of the detail in the figure is in fact captured in the main paragraph preceding the figure.
  • It is not recommended to provide graphs or figures in the conclusions, so I suggest to move “Figure 8” to the Results chapter, with comments on the information in that figure as well. Response: The mentioned figure (Figure 8 which now is Figure 5) is moved to the discussion section together with the accompanying texts. This provided the right place for it and we greatly appreciate the suggestion from the reviewer.  
  • The bibliographic descriptions of many information sources in the chapter “7 References” need to meet the requirements for authors. Response: All references are edited as per the guidance provided by the journal both in text and in the references section. It is now in full compliance with the journal references fomating style.

Reviewer 2 Report

The paper has some interesting observations and recommendations for ensuring sustainable use of natural resources within the context of traditional energy use in Africa. It seems that some of the projections on how much land can be restored and the degree to which biomass might be harvested from these lands are a bit optimistic, the paper is a useful starting point for discussion and offers some interesting discussion points.

The English could be improved so that the author’s meaning could be clarified at points. I have made comments below that might make the manuscript more readable. These are just a few comments but perhaps they might make the manuscript more readable.

First sentence in abstract is a little unclear. I think this makes the meaning clearer: “Energy supply systems in the tropics and subtropics are marred by considerable negative consequences on ecosystems.”

Line 22: replace “suffice about 2.5 billion people…” with “and supply sufficient energy for about 2.5 billion people..”.

Line 23 is a bit verbose, and the meaning is not precise. I’d prefer something along the lines of: “Ecosystem-based approaches to bioenergy along with a well-balanced involvement of industry coupled with knowledgeable management of the ecosystem could lead to beneficial outcomes for society.”

Line 37, change has to has. Change 1st “externalities” to consequences, 2nd  “externalities” to “negative effects” or similar.

Line 45, change “Besides” to “For example”

Line 47, change “who were using electricity” to “that use electricity”. On line 49, I don’t think either instance of “whole” is necessary.

Lines 50-52: Such are the characteristics of degenerating systems, where the elements of the system are leading to a vicious cycle of energy poverty and ecosystem degradation and could not lead to a breakthrough in themselves [3, 4]. Could changing this to … “and further complicating finding solutions to the problem.”

Line 65, delete the comma.

Line 72, add the words and disadvantages after benefits.  

Line 77, does the use of the word “externalities” here mean external inputs of energy? If so, would this sentence be clearer is it read something like “By examining and understanding the role ecosystems play in energy supply, we try to explore options for regenerative energy supply models by reducing reliance on external inputs such as coal and natural gas into the local energy supply and investing in unused local bioenergy resources along with judicial management of the local ecosystems.” I think this wording would be clearer to readers.

Line 80, change “will still dominate the space.” to “will still predominate.”. Also, the words “ improving the technological” are unnecessary, change to “besides technological advances..” Line 83 add “input” after “fully converted”.

Line 105, delete the word built. It is not necessary. Line 109, delete “it a”.

Line 118, Change the sentence “Among the key arguments in Regenerative economy [19]; is that wealth should be holistic and not the monetary value; hence, should include economic, social and environmental wealth. To: Among the key arguments in Regenerative economy [19]; is that wealth should be considered in a holistic manner and not simply as monetary value; hence, should include economic, social and environmental wealth.”

Line 124, changed speared to separate.

Line 236, change was to is.

Line 258, change to “Data for biomass resources was obtained from a report on forest products from The United Nations’ Food and Agriculture Organization [35].” For the next two references, mention the name of the source. Line 268, change to “which currently has no vegetation cover”.

It is not clear in section 3.3 but is this land that used to have some degree of vegetation cover, is there any historical data for estimating the degree of vegetation cover in the past?

Line 276, place period at end of sentence, delete the left parenthesis, and insert line before table 1.

Line 283, “We derived dry biomass equivalents for the emission values presented in [39].” Change instances such as these (as per the comment for line 258), by stating the author’s or organization’s name then the reference number.

Line 308 (and throughout manuscript), sufficed seems an awkward word, considered changing to something like this: “To analyse the potential number of people that can be adequately supplied with energy from this biomass,…”

Line 385, delete first comma.

Line 425, change to “When hydropower generation is limited due to droughts or dry seasons, the need..”.

As noted before, and here in line 431, and table 2, the word externalities is not very helpful,. In both of these two cases I think the word consequences would be much better.

Line 535, change “had” to “has”. I believe this statement should have a citation(s).

Line 581, The meaning of the sentence “Due to the interlinkages and consequently direct and inverse impact, a landscape thinking is required within policy development.” Is a bit unclear, should the word indirect be used instead of inverse?

Author Response

The paper has some interesting observations and recommendations for ensuring sustainable use of natural resources within the context of traditional energy use in Africa. It seems that some of the projections on how much land can be restored and the degree to which biomass might be harvested from these lands are a bit optimistic, the paper is a useful starting point for discussion and offers some interesting discussion points.

The English could be improved so that the author’s meaning could be clarified at points. I have made comments below that might make the manuscript more readable. These are just a few comments but perhaps they might make the manuscript more readable.

Response: Thank you for the very valuable comments on the document.  The document has been edited by a professional and a native speaker. A lot of edits were made. 

Specific comments

First sentence in abstract is a little unclear. I think this makes the meaning clearer: “Energy supply systems in the tropics and subtropics are marred by considerable negative consequences on ecosystems.”

Response: the said sentence is now revised and reads as ‘Energy supply systems in the tropics and subtropics are marred with considerable negative impacts on ecosystems, for example, forest loss and habitat destruction.’ With even more specific example.  See line 10-11.

Line 22: replace “suffice about 2.5 billion people…” with “and supply sufficient energy for about 2.5 billion people.”.

Response: Phrase now included, thanks

Line 23 is a bit verbose, and the meaning is not precise. I’d prefer something along the lines of: “Ecosystem-based approaches to bioenergy along with a well-balanced involvement of industry coupled with knowledgeable management of the ecosystem could lead to beneficial outcomes for society.”

Response: Phrase now included

Line 37, change has to has. Change 1st “externalities” to consequences, 2nd “externalities” to “negative effects” or similar.

Response: Phrase now changed

Line 45, change “Besides” to “For example”

Response: Phrase now changed

Line 47, change “who were using electricity” to “that use electricity”. On line 49, I don’t think either instance of “whole” is necessary.

Response: Phrases changed

Lines 50-52: Such are the characteristics of degenerating systems, where the elements of the system are leading to a vicious cycle of energy poverty and ecosystem degradation and could not lead to a breakthrough in themselves [3, 4]. Could changing this to … “and further complicating finding solutions to the problem.”

Response: Phrase now changed as suggested.

Line 65, delete the comma.

Response: Comma deleted

Line 72, add the words and disadvantages after benefits.

Response: change effected

Line 77, does the use of the word “externalities” here mean external inputs of energy? If so, would this sentence be clearer is it read something like “By examining and understanding the role ecosystems play in energy supply, we try to explore options for regenerative energy supply models by reducing reliance on external inputs such as coal and natural gas into the local energy supply and investing in unused local bioenergy resources along with judicial management of the local ecosystems.” I think this wording would be clearer to readers.

Response:  The word ‘externality’ repetitively used in the document is now replaced with simpler words such as consequences or impacts. It is broadly edited out. In relation to the specific sentence the reviewer pointed out, the sentence has been rephrased and broken into 2 --‘Here we examine and elucidate the role that ecosystems play in energy supply. We then explore options for regenerative energy supply models that reduce impacts of the energy supply system and invest in the management of the ecosystems while making use of unused resources.’ See Lines 80-82

Line 80, change “will still dominate the space.” to “will still predominate.”. Also, the words “ improving the technological” are unnecessary, change to “besides technological advances..” Line 83 add “input” after “fully converted”.

Response: suggested changes are corrected.

Line 105, delete the word built. It is not necessary. Line 109, delete “it a”.

Response: suggested changes are corrected.

Line 118, Change the sentence “Among the key arguments in Regenerative economy [19]; is that wealth should be holistic and not the monetary value; hence, should include economic, social and environmental wealth. To: Among the key arguments in Regenerative economy [19]; is that wealth should be considered in a holistic manner and not simply as monetary value; hence, should include economic, social and environmental wealth.”

Response: suggested changes are corrected. Thank you.

Line 124, changed speared to separate.

Response: suggested changes are corrected.

Line 236, change was to is.

Response: suggested changes are corrected.

Line 258, change to “Data for biomass resources was obtained from a report on forest products from The United Nations’ Food and Agriculture Organization [35].” For the next two references, mention the name of the source. Line 268, change to “which currently has no vegetation cover”.

Response: All suggested changes made and both names of the references are mentioned in the revised document.

It is not clear in section 3.3 but is this land that used to have some degree of vegetation cover, is there any historical data for estimating the degree of vegetation cover in the past?

Response: It is an area of land with vegetation of less 10% cover. The way it is derived in the main data base provided by FAO is that it used to be a forest land or savannah and its gradually degraded to be degraded vegetated area. With careful indepth analysis, it is possibel to know but since it goes beyond the scope of this analysis we did not include it here. 

Line 276, place period at end of sentence, delete the left parenthesis, and insert line before table 1.

Response: suggested change made and corrected.

Line 283, “We derived dry biomass equivalents for the emission values presented in [39].” Change instances such as these (as per the comment for line 258), by stating the author’s or organization’s name then the reference number.

Response: this is corrected. Author name now appears first.

Line 308 (and throughout manuscript), sufficed seems an awkward word, considered changing to something like this: “To analyse the potential number of people that can be adequately supplied with energy from this biomass,…” .

Response: The word suffice is replaced fully in the text. Thank you.

Line 385, delete first comma.

Response: suggested change made and corrected.

Line 425, change to “When hydropower generation is limited due to droughts or dry seasons, the need..”. 3

Response: suggested change made and corrected.

As noted before, and here in line 431, and table 2, the word externalities is not very helpful,. In both of these two cases I think the word consequences would be much better.

The word externality has been replaced throughout the text. Hence corrected.

Line 535, change “had” to “has”. I believe this statement should have a citation(s).

Response: suggested change made and corrected. We have added two references to it as suggested. See lines 536-537 in the revised document.  

Line 581, The meaning of the sentence “Due to the interlinkages and consequently direct and inverse impact, a landscape thinking is required within policy development.” Is a bit unclear, should the word indirect be used instead of inverse?

Response: suggested change made and corrected. Sorry for using a vague word here. See lines 581-582.

Reviewer 3 Report

The main contribution of this work is to evaluate bioenergy generation by building on the regenerative economy concept, based on the analysis of ecosystems in household energy supply in Africa. This topic is interesting and meaningful. It can provides a good reference for future  ecosystem-based approaches to bioenergy. Some concerns listed as follows: 

  1. Graphical abstract should be added in the paper, especially as a high quality paper which will be published in sustainability.
  2. In Abstract, line 10, The author wrote that “Energy supply systems in the tropics and subtropics are marred with considerable negative externalities on ecosystems”. Could you listed some samples of negative externalities on ecosystems? This brief information should be listed in Abstract to make reader clear.
  3. In introduction, line 79, what is meaning of “EIA”?
  4. In this paper, authors focus on the perspectives of biomass energy. However, low conversion rates and poor economic benefits remain the biggest problems in biomass applications. Authors should consider the availability of biomass energy. For example, biomass energy is widely used in Europe, but has encountered great difficulties in Asia, why? This part need to be considered and described in the article.

Author Response

Comments and Suggestions for Authors

The main contribution of this work is to evaluate bioenergy generation by building on the regenerative economy concept, based on the analysis of ecosystems in household energy supply in Africa. This topic is interesting and meaningful. It can provides a good reference for future  ecosystem-based approaches to bioenergy. Some concerns listed as follows: 

General repsonse: we greatly appreciate the comments and we hope so too. For so long the discussion on energy remained mono-sectoral. This kind of research will stir the discussion around how we can build the energy issues on a system’s level connectivity and understanding. The language for the manuscript is editted by a professional and a native speaker. 

Graphical abstract should be added in the paper, especially as a high quality paper which will be published in sustainability.

Response: Graphical abstract is included in this manuscript now. Thank you for the suggestion. See page 2.

In Abstract, line 10, The author wrote that “Energy supply systems in the tropics and subtropics are marred with considerable negative externalities on ecosystems”. Could you listed some samples of negative externalities on ecosystems? This brief information should be listed in Abstract to make reader clear.

Response: The sentence is revised as ‘Energy supply systems in the tropics and subtropics are marred with considerable negative impacts on ecosystems, for example, forest loss and habitat destruction.” The examples are also included as indicated.  See lines 10-11.

In introduction, line 79, what is meaning of “EIA”?

Response: Full name included ‘The U.S. Energy Information Administration (EIA)’. Thank you. See line 82-83.

In this paper, authors focus on the perspectives of biomass energy. However, low conversion rates and poor economic benefits remain the biggest problems in biomass applications. Authors should consider the availability of biomass energy. For example, biomass energy is widely used in Europe, but has encountered great difficulties in Asia, why? This part need to be considered and described in the article.

Response: We have added a whole new paragraph (described below) the deliberated the point raised by the reviewer. See lines 641-653 in the revised documents.

“Low adoption of ecosystem-based bioenergy interventions is associated with inadequate enabling national policy and regulatory framework needed to attract commercially viable investment, especially in Africa [78]. Despite its enormous potentials, the big question is often - who would be willing to invest in such schemes? Technical and technological weaknesses, including trained personnel, necessary equipment and infrastructure, are a major hinderance as Luthra et al. [79] notes. Most of these technologies are not locally available, thus attracting high importation, installation and maintenance costs and consequently low adoption. There is also a general lack of incentives such as credit facilities and subsidies in many African countries to enhance more investments in these interventions despite their huge potential to address energy demands. From the marketability perspective, some of the key barriers are high costs of bioenergy production and service delivery, competition from traditional non-renewable sources such as fossil fuels, poor market development of ecosystem-bioenergy sources and the finite (non-renewable) nature of bioenergy raw materials [80], which have as a result reduced their adoption rates.”

Round 2

Reviewer 3 Report

The author's feedback addressed all my concerns. I think it can be accepted by Sustainability now.